# Study of the Natural Crystalline Lens Characteristics Using Dual-Energy Computed Tomography

**DOI:** 10.3390/diagnostics12112857

**Published:** 2022-11-18

**Authors:** Jeffrey R. Sachs, Javier A. Nahmias, Kevin D. Hiatt, James G. Bomar, Thomas G. West, Paul M. Bunch, Marc D. Benayoun, Chris Lack, Atalie C. Thompson

**Affiliations:** 1Department of Radiology, Atrium Health Wake Forest Baptist Medical Center, 1 Medical Center Boulevard, Winston-Salem, NC 27157, USA; 2Department of Surgical Ophthalmology, Atrium Health Wake Forest Baptist Medical Center, 1 Medical Center Boulevard, Winston-Salem, NC 27157, USA

**Keywords:** dual-energy computed tomography, natural crystalline lens, sex, race

## Abstract

There is a paucity of radiologic literature regarding age-related cataract, and little is known about any differences in the imaging appearance of the natural crystalline lens on computed tomography (CT) exams among different demographic groups. In this retrospective review of 198 eyes in 103 adults who underwent dual-energy computed tomography (DECT) exams of the head, regions of interest spanning 3–5 mm were placed over the center of the lens, and the x-ray attenuation of each lens was recorded in Hounsfield Units (HU) at 3 energy levels: 40 keV, 70 keV, and 190 keV. Generalized estimating equations (GEEs) were used to assess the association of clinical or demographic data with lens attenuation. The mean HU values were significantly lower for the older vs. younger group at 40 keV (GEE *p*-value = 0.022), but there was no significant difference at higher energy levels (*p* > 0.05). Mean HU values were significantly higher for females vs. males and non-whites vs. non-Hispanic whites at all 3 energy levels in bivariate and multivariable analyses (all *p*-value < 0.05). There was no significant association between lens attenuation and either diabetes or smoking status. The crystalline lens of females and non-whites had higher attenuation on DECT which may suggest higher density or increased concentration of materials like calcium and increased potential for cataract formation. Given the large scope of cataracts as a cause of visual impairment and the racial disparities that exist in its detection and treatment, further investigation into the role of opportunistic imaging to detect cataract formation is warranted.

## 1. Introduction

Cataracts are the leading cause of blindness worldwide and affect over 24 million Americans [1]. Of the subtypes of age-related cataracts, nuclear sclerotic cataracts are the most common [2], and are thought to result from cumulative lifetime exposure to a range of insults to the ocular lens including ultraviolet light, ocular trauma, ocular surgery, corticosteroid use, radiation exposure, smoking, and diabetes mellitus.

Currently, cataracts are almost universally a clinical diagnosis, made by confirming opacification or discoloration of the lens during slit lamp examination by a trained specialist in optometry or ophthalmology. However, barriers to access of basic healthcare services have caused many patients to seek acute medical care through the emergency room, which provides an opportunity for incidental discovery of comorbid chronic conditions, such as cataracts [3].

Computed tomography (CT) scans are one of the most common medical tests performed in the emergency room [4]. CT has proven useful in diagnosing traumatic cataracts [5,6,7,8], wherein the lens becomes hypoattenuating due to increased fluid content. However, there is a striking paucity of radiologic literature on the imaging characteristics of sage-related cataracts, which are far more common than traumatic cataracts [2]. Moreover, little is known about differences in the characteristics of the natural lens on CT imaging among different demographic groups.

One reason for this lack of knowledge is that when using conventional single-energy CT techniques, the lenses are bland, relatively homogeneous structures that garner little attention unless they are displaced, dysmorphic, or absent. Moreover, since the natural lens is a radiosensitive organ at particular risk for radiation effects due to its superficial location in the human body, intentional imaging of the lens during radiologic image acquisition is not generally recommended [9]. In clinical practice, however, the lens is often incidentally included in the CT scan field of view. This circumstance presents the radiologist with an opportunity to assess the lens for pathology such as cataracts.

In recent years, dual-energy CT (DECT) has risen to the forefront of CT imaging acquisition due to its ability to capitalize on the differences in energy-dependent x-ray absorption of different materials within the patient by using low- and high-energy x-ray spectra. DECT techniques have numerous recently described applications in the practice of neuroradiology [10]. Low kilo–electron volt (keV) (e.g., 40 keV) virtual monoenergetic imaging (VMI) techniques allow for improved contrast-to-noise ratio among soft tissues of similar attenuation, even in the absence of iodinated contrast media [11]. High keV (e.g., 190 keV) VMI have been used to describe unique attenuation patterns of silicone oil which can be found in the eye after certain retinal surgeries [12,13].

Given the ability of DECT to improve discrimination between a material of high atomic number (such as calcium or iodine) and a material of low atomic number (such as hemorrhage) [14], we hypothesized that it would be useful in detecting differences in lens composition that could be related to early cataract formation. The lens is made of proteins which are low in atomic number. If the cataractic lens were to calcify, then one would expect that the attenuation of the lens would increase fairly dramatically over time, especially at 40 keV, due to the combined impact of low energy incident photons and high atomic number on the photoelectric effect, and subsequently on x-ray beam attenuation. Although total lens calcification is rare [15], lens calcium content has been shown to correlate with the degree of opacification of cataractous lenses [16]. Impaired intracellular Ca+2 signaling in lens epithelial cells is known to play an intrinsic role in both cortical and nuclear cataractogenesis [17]. Moreover, we might expect nuclear cataracts to have an increased density due to the pathologic aggregation and compaction of the proteins of the nuclear fibers which has been observed on histology [2]. However, to our knowledge, no study has investigated the characteristics of the natural crystalline lens on DECT or whether there are differences in attenuation of the lens in different demographic groups or at different levels of energy.

In this retrospective study, we examined whether there is a relationship between possible demographic and clinical risk factors for cataract formation, such as age, race, sex, diabetes and smoking status, with the X-ray attenuation of the natural crystalline lens on DECT of the head at 40, 70, and 190 keV VMI. Such imaging analyses of the crystalline lens may provide critical foundational understanding of the changes in the lens that one might observe on DECT as well as help identify groups at potential risk for cataract formation.

## 2. Materials and Methods

### 2.1. Subjects

For this retrospective, Health Insurance Portability and Accountability Act-compliant, institutional review board-approved study, a local institutional radiology database (syngo.via, Siemens Healthineers) was queried with the goal of identifying eligible lenses for study inclusion. Screening eligibility criteria were (1) age ≥18 years and (2) received non-contrast DECT of the head at our institution between July and December 2020. A younger adult (age ≥18 and <30 years) and older adult cohort (age >70 years) were collected to facilitate assessment of any age-related differences in the lens characteristics. Patients were excluded if there was evidence of acute orbital trauma (including periorbital hematoma, orbital hemorrhage, and orbital fractures). Pseudophakic eyes were not included in the analysis. Using these criteria, a total of 198 eyes in 103 adult subjects were identified. Demographic or clinical risk factors for cataract such as age, sex, race/ethnicity, smoking status, and history of diabetes were collected from the electronic medical record. Documentation of known cataract status was also collected when available.

### 2.2. Image Acquisition

All non-contrast DECT examinations of the head were performed utilizing a dual-source system (SOMATOM Drive or Flash; Siemens Healthineers, Erlangen, Germany). The DECT acquisition parameters were as follows: 80 kVp/Sn140 kVp acquisitions, Quality Reference mAs of 400/200, pitch of 0.70, rotation time of 0.5 s, and with automated tube current modulation, CareDose 4D.

### 2.3. Image Analysis

All dual-energy post-processing and region of interest (ROI) analysis was performed in syngo.via (Siemens Healthineers) using the Monoenergetic+ application. Measurements were obtained across all subjects using an image slice thickness of 1 mm. Two-dimensional circular ROIs were drawn by a trained radiology resident or neuroradiology fellow in the center of the native lenses. Care was taken to identify the center of the lens and avoid any streak or beam hardening artifacts. An attending neuroradiologist with 5 years subspecialty experience reviewed and optimized the ROI position as needed. All ROIs were sized to be between 3–5 mm^2^. A representative example of how ROI measurements were obtained is given in Figure 1. After the initial ROI was placed, the ROI location did not change while HU attenuation measurements were recorded at each of the 40 keV, 70 keV, and 190 keV energy levels. The mean attenuation and standard deviation (SD) were recorded in Hounsfield Units (HU) for each ROI at each energy level.

### 2.4. Statistical Analyses

Descriptive statistics were used to describe the clinical and demographic characteristics of the population. Pearson’s correlation was used to compare the attenuation in Hounsfield units between right and left eyes among subjects with both natural lenses present on DECT. In order to account for the correlation of both eyes within a given subject, separate generalized estimating equations (GEEs) were constructed to assess the bivariate relationship between each of the demographic or clinical variables and the relative attenuation of the lens at each energy level (40, 70, and 190 keV). Factors that were significant in bivariate analyses (*p*-value < 0.05) were entered into multivariable GEE models of the HU for each energy level. All analyses were performed with Stata (version 17.0, StataCorp, College Station, TX, USA). A *p*-value < 0.05 was considered statistically significant.

## 3. Results

### 3.1. Subjects

Included subjects were sub-categorized into an older adult (N = 53; mean age 81.4 ± 5.7 years; range 73–101) and younger adult cohort (N = 50; mean age 22.66 ± 2.93 years; range 18–27). Approximately 46.6% of subjects were female, 38.8% self-identified as non-Hispanic white, 22.3% had diabetes, and 37.9% were never smokers (Table 1). Among the older adults, only 19 subjects (29 eyes) had ophthalmic examination data of the natural lens in their electronic healthcare record so that cataract status could be determined.

### 3.2. Image Analysis

Ninety-five subjects had a natural crystalline lens present in both eyes. There was a strong and significant positive correlation between the HU attenuation values in the right and left eye at each of the three energy levels (all *p* < 0.0001) (Table 2).

Table 3 displays the bivariate association between clinical and demographic characteristics and attenuation of the natural crystalline lens in the full cohort (198 eyes in 103 subjects). At 40 keV, the older adults had a significantly lower average HU than younger adults (70.88 ± 13.63 vs. 75.70 ± 13.21; *p* = 0.020), but there was no significant difference in the measurements for older vs. younger adults at the 70 keV or 190 keV energy levels (both *p* > 0.05). However, in the subgroup of older adults with clinical documentation of a cataract (N = 19 subjects with 29 eyes), the Hounsfield unit values were higher, measuring 74.65 ± 14.57) at 40 keV, 79.05 ± 7.58 at 70 keV, and 80.85 ± 9.29 at 190 keV.

At all three energy levels, the mean HU attenuation values were significantly higher for females vs. males (40 keV: 76.4 ± 13.3 vs. 70.7 ± 13.4; 70 keV: 79.0 ± 7.4 vs. 74.2 ± 7.6; 190 keV: 80 ± 7.9 vs. 75.8 ± 9; all *p* <= 0.01) and for non-whites vs. non-Hispanic whites (40 keV: 77.6 ± 13.1 vs. 70.5 ± 13.2; 70 keV: 80 ± 7.9 vs. 74.1 ± 7.0; 190 keV: 81.0 ± 8.1 vs. 75.5 ± 8.5; all *p* <= 0.001). No significant association was detected between lens attenuation and either diabetes or smoking status (all *p* > 0.05).

In multivariable analyses, both female sex (all *p* < 0.01) and nonwhite race/ethnicity (all *p* < 0.01) but not age-group (*p* > 0.05) remained significant independent predictors of lens attenuation at all 3 energy levels (Table 4).

## 4. Discussion

To our knowledge, this study is the first to demonstrate the ability of DECT to detect differences in the attenuation of the crystalline lens on DECT among demographic and clinical groups that may be at increased risk for cataract formation.

We found that the crystalline lens of females and non-white subjects had significantly higher attenuation on DECT at all energy levels. This finding may suggest a higher density lens or increased concentration of materials like calcium which could be related to lens opacification from early cataract formation [16]. The Salisbury Eye Evaluation Study found that African Americans had higher rates of cortical opacity and progression of cortical cataracts [18]. The Age-Related Eye Disease study similarly found that both females and non-whites were at greater risk of cortical cataract formation [19]. One hypothesis in females is that age-related withdrawal of estrogen may play a role in the progression cataracts through the loss of its anti-oxidative effects [20]. Women also have a higher prevalence of osteoporosis which has been associated with increased risk for cataract, presumably through common pathways of impaired calcium homeostasis [21]. Such impaired calcium signaling in lens epithelial cells can result in increased cytosolic calcium concentration [22] which predisposes to cortical cataracts in particular. For example, one study demonstrated that the total calcium in lenses with cortical cataracts measured four times higher than in clear lenses [23]. Cortical cataracts can also have discrete calcium deposits which would have substantially higher attenuation on DECT. Moreover, impaired calcium signaling can also lead to opacification of the nuclear portion of the lens resulting in a mixed cataract type [17]. Interestingly, the subset of older adults with a documented cataract also had higher HU values. However, the retrospective design of the study limited more direct assessment of the relationship between these characteristics and cataract formation since ophthalmic examination data was not available in a majority of patients. Nevertheless, the study demonstrated that DECT has the potential to identify differences in lens attenuation among groups at risk for cataract. Such findings could form the rationale for a future prospective study in which patients undergoing DECT of the head are recruited for ophthalmology examination to determine whether a clinically and visually significant opacification of the lens is present.

To our surprise, no significant association between lens attenuation and either diabetes or smoking status was detected (all *p* > 0.05). High blood glucose in the setting of poorly controlled diabetes can lead to generation of polyols that result in increased osmotic stress in the lens fibers causing them to swell and rupture [24]. Given previous studies on traumatic cataracts and the low HU attenuation from increased water concentration, one might have expected diabetic patients to have a lower HU attenuation. The lack of an association here could be related to the fact that severity of diabetes was not able to be determined since HbA1c was not consistently available. Future studies could collect additional lab criteria to stratify by diabetic control. Similarly, prior studies have associated smoking with an increased risk of nuclear sclerotic type cataracts due to increased oxidative stress [25,26]. It is possible that smoking data from the medical record is less accurate than questionnaire data which could be collected in a future prospective study.

The older adult cohort in general trended toward a lower attenuation of the lens at 40 keV, but this relationship was not significant in multivariable analyses or at higher energy levels. It is possible that DECT did not identify a Hounsfield-unit based threshold to distinguish lens age because different subtypes of cataract may result in different alterations in lens attenuation. For example, some cataracts might decrease lens attenuation if there is an increase in fluid whereas others may increase lens attenuation if there is an increase in deposition of calcium or other high-density materials. Future studies could consider whether specific subtypes of cataract have different degrees of lens attenuation on DECT.

Although an ROI based methodology to confirm cataract by DECT is not yet possible, the lens remains a potentially attractive target for artificial intelligence-based segmentation given its simple shape, near-uniform size, and the high level of contrast between the lens itself and the surrounding fluid of the aqueous and vitreous humor [27]. Larger DECT imaging datasets pooled across multiple institutions may make possible the training and development of deep learning algorithms for predicting cataract risk. Additionally, while obtaining the study data we noticed occasional patients that demonstrated highly variable attenuation values in the lenses that produced a “speckled” pattern of attenuation (Figure 2). Such a pattern suggests the usefulness of texture analysis for identification of cataracts, where differences in spatial heterogeneity within the ROI of a cataract lens can be distinguished from a non-cataract lens, despite having, for example, similar overall mean HU attenuation values. In fact, prior studies have reported greater than 90% accuracy in identifying cataracts from eye photographs when using lens ROI uniformity (a texture analysis metric) as a training feature for a nearest-neighbor classifier [28], though to the best of our knowledge this has not yet been reproduced with DECT images.

A majority of patients in our study did not have historic ophthalmic data, suggesting there could be gaps in access to ophthalmic care that could be potentially addressed during an emergency department visit. Racial disparities in access to cataract surgery are well-documented, especially among black patients [29]. Inequities in the proportion of patients that carry health insurance affects access to routine health care and utilization [30,31]. Racial minorities are also known to be less likely than white patients to have a primary care provider and are more likely to rely on the Emergency Department for routine care needs [32,33,34]. As such, radiologists could have an opportunity to help improve healthcare disparities through invention of novel opportunistic imaging screening techniques (i.e., screening an organ for pathology when that organ is incidentally imaged as part of a study obtained for a separate indication). While traditional cataract evaluation will remain the gold-standard for cataract diagnosis, future studies should investigate whether patients presenting to the ED who undergo DE-CT could benefit from opportunistic detection of cataract and development of a care pathway that refers such patients to ophthalmology for confirmation and treatment.

Our study has several limitations. In order to allow for examination of possible age-related differences in the lens, two cohorts of older and younger adults were collected. Thus, the study findings are not generalizable to middle-aged patients. Achieving accurate lens cortex attenuation values is problematic due to the concern for volume averaging with adjacent fluid filled structures. However, if eyes had cortical changes overlying the nucleus these would have been detected within the ROI. Additional ophthalmic clinical data as to the presence or grade of cataract were only available in a subset of older adults. Based on the findings in this study, a future prospective study could be designed in which patients are referred to an ophthalmologist for clinical exam of the lens following incidental imaging of the crystalline lens on DECT during an emergency room encounter.

Another potential limitation of the study is the presence of beam hardening artifacts at 40 keV VMI. In our study, the authors took great care to avoid beam hardening artifacts emanating from the bony orbit; if such artifacts had been included in the ROI analysis, they would have led to spurious increases or decreases in HU values. Photon counting CT holds promise to both improve spectral resolution and reduce beam hardening artifacts in the future [35].

Our study was performed on one of two distinct types of DECT scanners, both dual-source systems manufactured by Siemens, and with nearly identical protocols. These results may not necessarily be generalizable to all commercially available forms of spectral CT imaging, though the underlying physics should remain the same. Moreover, several factors can affect the accuracy of the HU values, including the aforementioned beam hardening artifacts, spectral energy, convolution kernel, and patient positioning [36]. CT scanners are calibrated such that the HU value for pure water does not deviate more than 2 HU from the reference value of 0 [37], but the possibility exists that small differences in scanner calibration in conjunction with a combination of the above factors could impact the measurements taken.

## 5. Conclusions

Our study found that non-white and female patients had significantly higher lens attenuation values on DECT, which may suggest increased lens density, the presence of high-density materials like calcium, or a predisposition for cataracts. Future collection of ophthalmic examination data in patients with DECT findings will provide useful diagnostic confirmation of the clinical relevance of these findings. Given the large scope of cataracts as a cause of visual impairment and the racial disparities that exist in its detection and treatment, further investigation into the role of opportunistic DECT imaging to detect cataract formation is warranted. As these technologies continue to advance, incorporation of photon counting CT and artificial intelligence-based textural analysis may improve the utility of such radiologic studies in the clinical care of patients with incidental cataracts.

## Figures and Tables

**Figure 1 diagnostics-12-02857-f001:**
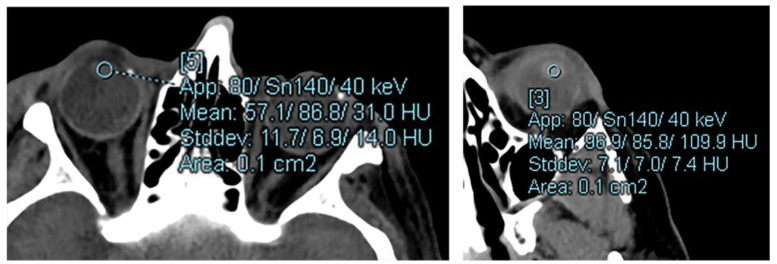
A representative example of how ROI measurements were obtained for the study. All measurements were obtained in the axial plane using 1 mm thick slices. ROI data was obtained using the default syngo.via Monoenergetic+ application. All scans were non-contrast dual-source, dual-energy CT exams. See the text Section 2.2. for the DECT acquisition parameters.

**Figure 2 diagnostics-12-02857-f002:**
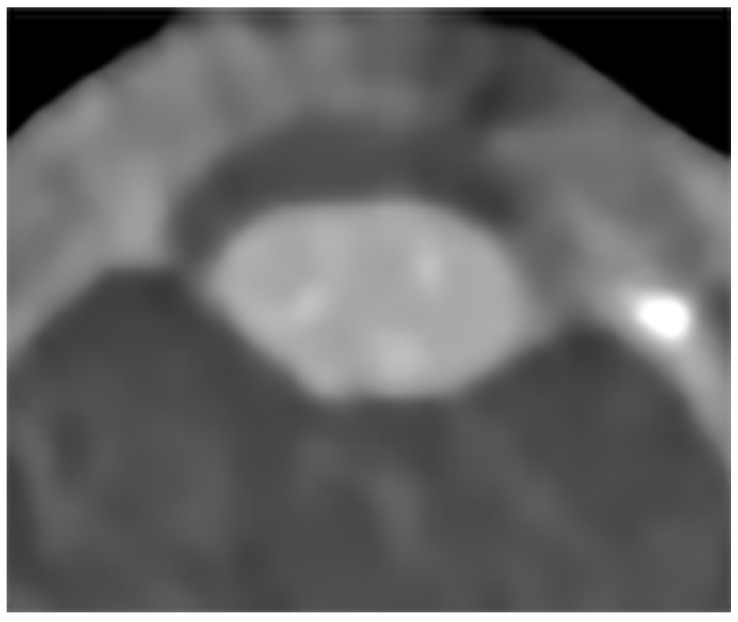
Axial non-contrast DECT image zoomed for lens detail. An 84-year-old lens demonstrates a “speckled” appearance—with scattered punctate foci of increased attenuation. This pattern of spatial heterogeneity raises the question of whether texture analysis may be a more useful method of assessing for cataract at CT.

**Table 1 diagnostics-12-02857-t001:** Clinical and demographic characteristics of 103 adult subjects who underwent Dual Energy Computed Tomography (DE-CT).

Characteristics	N (%)
**Age**	
Older Adults (73–101 years)	53 (51.5%)
Young Adults (18–27 years)	50 (48.5%)
**Sex**	
Female	48 (46.6%)
Male	55 (53.4%)
**Race/Ethnicity**	
Non-White	40 (38.8%)
Non-Hispanic White	63 (61.2%)
**Smoking Status**	
Ever Smoker	39 (37.9%)
Never-smoker	64 (62.1%)
**Diabetes**	
Yes	23 (22.3%)
No	80 (77.7%)

**Table 2 diagnostics-12-02857-t002:** Correlation of right and left eye attenuation measurements on Dual-Energy Computed Tomography in 95 pairs of eyes in 95 subjects.

	Mean ± Standard Deviation, Hounsfeld Units in keV	Mean ± Standard Deviation, Hounsfeld Units in keV		
**DECT** **Energy Level**	Right Eye	LEFT EYE	Pearson’s Correlation	*p*-Value
40 keV	73.5 ± 12.6	74.0 ± 14.1	0.4061	**<0.0001**
70 keV	76.3 ± 8.0	76.6 ± 7.9	0.6706	**<0.0001**
190 keV	77.5 ± 8.7	77.7 ± 8.8	0.6926	**<0.0001**

DECT = Dual-Energy Computed Tomography. Bolded *p*-values are statistically significant (*p* < 0.05).

**Table 3 diagnostics-12-02857-t003:** Bivariate analysis of clinical and demographic characteristics associated with natural crystalline lens attenuation measured at three energy levels on Dual Energy Computed Tomography (DE-CT) (N = 198 eyes in 103 subjects).

Characteristics	Hounsfield Units (HU) in keV		
	40 keV	70 keV	190 keV
**Age**			
Older Adults	70.88 ± 13.63	75.18 ± 7.77	76.91 ± 9.26
Young Adults	75.70 ± 13.21	77.64 ± 7.78	78.48 ± 8.17
GEE *p*-value	**0.022**	0.080	0.381
**Sex**			
Female	76.41 ± 13.28	79.0 ± 7.37	79.99 ± 7.91
Male	70.68 ± 13.37	74.24 ± 7.61	75.76 ± 8.97
GEE *p*-value	**0.010**	**<0.0001**	**0.005**
**Race/Ethnicity**			
Non-White	77.63 ± 13.12	79.98 ± 7.86	81.03 ± 8.12
Non-Hispanic White	70.51 ± 13.22	74.11 ± 6.96	75.54 ± 8.48
GEE *p*-value	**0.001**	**<0.0001**	**0.001**
**Smoking Status**			
Ever Smoker	71.81 ± 14.92	75.30 ± 8.04	76.69 ± 9.71
Non-smoker	74.23 ± 12.71	77.12 ± 7.68	78.32 ± 8.07
GEE *p*-value	0.275	0.224	0.361
**Diabetes**			
Yes	70.83 ± 13.01	75.96 ± 6.88	77.98 ± 9.59
No	73.98 ± 13.72	76.55 ± 8.11	77.63 ± 8.53
GEE *p*-value	0.168	0.810	0.588

Generalized estimating equations = GEE. Bolded *p*-values are statistically significant (*p* < 0.05).

**Table 4 diagnostics-12-02857-t004:** Multivariable generalized estimating equations of characteristics associated with natural crystalline lens attenuation measured at three energy levels on Dual Energy Computed Tomography (DE-CT) (N = 198 eyes in 103 subjects).

Characteristics	Hounsfield Units (HU) in keV		
	40 keV Beta (95% CI)	70 keV Beta (95% CI)	190 keV Beta (95% CI)
**Age**			
Older Adults	0.024 (0.0003, 1.63)	0.272 (0.022, 3.34)	0.789 (0.406, 15.33)
Young Adults	Reference	Reference	Reference
GEE *p*-value	0.083	0.309	0.876
**Sex**			
Female	374.5 (6.54, 21434.22)	124.42 (11.3, 1369.9)	71.06 (4.17, 1210.02)
Male	Reference	Reference	Reference
GEE *p*-value	**0.004**	**<0.0001**	**0.003**
**Race/Ethnicity**			
Non-White	403.47 (5.367, 30333.49)	208.07 (16.04, 2698.38)	179.92 (8.70, 3718.84)
Non-Hispanic White	Reference	Reference	Reference
GEE *p*-value	**0.006**	**<0.0001**	**0.001**

GEE = Generalized estimating equations; CI = confidence interval. Bolded *p*-values are statistically significant (*p* < 0.05).

## Data Availability

Restrictions apply to the availability of these data. Data were obtained from the electronic healthcare record at Atrium-Wake Forest Baptist Health and a de-identified dataset may be available with the permission of Atrium-Wake Forest Baptist Health’s legal team.

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
