# Peer review of "Study of the Natural Crystalline Lens Characteristics Using Dual-Energy Computed Tomography"

_diagnostics, 2022, doi:10.3390/diagnostics12112857_

Round 1
Reviewer 1 Report
1. This study did not differentiate between clear and cataractous lenses (and their extent of opacity). This leads to two results: (1) It cannot be determined whether this article is based on normal population (natural crystalline lens). (2) CT results cannot be correlated with the degree of lens opacity. For example, we still do not know whether the decrease in CT HU corresponds to the degree of cataract, therefore the article was lack of clinical significance.
2. As the authors mentioned, "since the natural lens is a radiosensitive organ at particular risk for radiation effects due to its superficial location in the human body, intentional imaging of the lens during radiologic image acquisition is not generally recommended."
On the other hand, the clinical diagnosis of lens opacities/cataracts is quite simple and convenient by direct or slit lamp observation without any additional radioactive or invasive examinations. This also decreased the clinical significance of this paper.
3. The last line on the first page: by “sage-related cataracts” I think the author means age-related cataracts.
Author Response
Diagnostics 11/4/22
Dear Ms. Wang,
We thank the editor and reviewers for their thoughtful reviews of our submission and the opportunity to revise the manuscript, with the now revised title “Study of the Natural Crystalline Lens Characteristics Using Dual-Energy Computed Tomography”. Please see our bolded responses to reviewer comments below. As we note, this paper is the first to explore the ability of Dual-Energy Computed Tomography (DECT) to detect differences in the attenuation of the natural crystalline lens between different demographic groups at risk for cataract. While we acknowledge there are limitations when working with a retrospective dataset, we believe the findings are novel and make a meaningful contribution to the field. Moreover, we hope these data will provide the basis of important future prospective work where more definitive ophthalmic evaluation can be proposed in patients undergoing DE-CT of the head during an ED visit. Given the novel imaging findings, we believe this work is well suited to publication in Diagnostics in the special issue Lesion Detection and Analysis using Optical Imaging. Please do not hesitate to reach out if any further questions or requests.
Sincerely,
Jeremy Sachs, MD and Atalie C. Thompson, MD, MPH
Response to Reviewers
Reviewer #1:
- This study did not differentiate between clear and cataractous lenses (and their extent of opacity). This leads to two results: (1) It cannot be determined whether this article is based on normal population (natural crystalline lens). (2) CT results cannot be correlated with the degree of lens opacity. For example, we still do not know whether the decrease in CT HU corresponds to the degree of cataract, therefore the article was lack of clinical significance.
We thank the reviewer for their comment. This paper is the first to explore the ability of Dual-Energy Computed Tomography (DECT) to detect differences in the attenuation of the natural crystalline lens between different demographic groups at risk for cataract. As a first study, it was a retrospective design that enrolled patients who had previously presented to the emergency department for DECT imaging of the head. Since the natural lens is opportunistically captured by this imaging, it presented the unique opportunity to examine the natural lens characteristics on DECT to see if there might be any discernible differences between patients of differing demographics. However, since this was a retrospective study, it presents limitations which we acknowledge in the paper; one of those limitations is that most patients did not have ophthalmology exam historic data available which limited our ability to directly compare ophthalmic lens gradings with measured HU on DECT. In the elderly cohort, formal ophthalmologic exam records were only available on a small sub-cohort, which we also analyzed in a limited capacity based on what we believed was feasible and reasonable. In addition, we did collect the available lens gradings from historic chart data, but we did not present analyses with these grades as they were not standardized, were often difficult to interpret presenting ranges of grades (i.e. 1-2+ NS, 3-4+ CS), and could be mixed including nuclear sclerosis, cortical, and posterior subcapsular types of cataract in the same lens. Ultimately clinical lens gradings are subjective, and more formal measurements of lens opacity are not performed in clinical practice so were not available but might be possible in a future prospective cohort study. Patients in the young cohort did not have historic ophthalmology exam data available in the medical record, but given their young age it is reasonable to assume they would have clearer crystalline lenses.
Ultimately, the aim of our study was not to compare cataracts with normal lenses on CT as this is not possible given the way these patient data were captured – i.e. retrospectively and opportunistically via DE-CT head scans performed during a prior emergency department visit. Rather, we aimed to explore whether there may be differences in HU on DE-CT for demographic and clinical risk factors that could be obtained in a retrospective chart review. These are the first ever analysis to consider the demographic differences in lens characteristics on DE-CT and the findings observed raise several important hypotheses which could be pursued in a future study. For this reason, we do believe this is a novel study that makes an important contribution to the literature. Based on these data, a future prospective study could be designed to recruit from a cohort of patients who are already undergoing DECT of the head and invite them to have subsequent ophthalmic examination (with formal measurements of lens opacity) as part of a funded clinical research trial. We have added this point to the discussion, as we do believe that these retrospective data provide the important foundation needed to propose such a prospective study.
Please see Discussion paragraph 2, page 6-7:
“Interestingly, the subset of older adults with a documented cataract also had higher HU values. However, the retrospective design of the study limited more direct assessment of the relationship between these characteristics and cataract formation since ophthalmic examination data was not available in a majority of patients. Nevertheless, the study demonstrated that DECT has the potential to identify differences in lens attenuation among groups at risk for cataract. Such findings could form the rationale for a future prospective study in which patients undergoing DECT of the head are recruited for ophthalmology examination to determine whether a clinically and visually significant opacification of the lens is present.”
- As the authors mentioned, "since the natural lens is a radiosensitive organ at particular risk for radiation effects due to its superficial location in the human body, intentional imaging of the lens during radiologic image acquisition is not generally recommended." On the other hand, the clinical diagnosis of lens opacities/cataracts is quite simple and convenient by direct or slit lamp observation without any additional radioactive or invasive examinations. This also decreased the clinical significance of this paper.
The authors agree that CT should not, and will not, replace the traditional method of diagnosing cataract by direct or slit lamp exam during ophthalmic exam. However, many patients do not have routine ophthalmic care (i.e. a majority of patients in this study had no historic ophthalmology visits), but may be seen in the emergency room and undergo imaging like DE-CT of the head that incidentally also images the natural lens. These data of the lens are incidentally captured, but currently are not analyzed or utilized, which is a potential missed opportunity. This retrospective analysis suggests that differences in the natural lens can be detected between different demographic groups, which suggests that further prospective investigation is worth pursuing to determine whether discrete diagnoses (like cataract) can be distinguished. However, this would require a prospective design as described above and these retrospective analyses do provide a foundation for such a future proposal. While these are exploratory and early findings, we believe they are promising. In the future if we could identify cataracts on DE-CT, there may be a role for opportunistic identification of cataract in patients already undergoing DE-CT of the head (for other reasons), with a care pathway for referral to ophthalmology to confirm and treat the diagnosis.
Please see modification to the Discussion, paragraph 5, page 8:
A majority of patients in our study did not have historic ophthalmic data, suggesting there could be gaps in access to ophthalmic care that could be potentially addressed during an emergency department visit. Racial disparities in access to cataract surgery are well-documented, especially among black patients. [30] Inequities in the proportion of patients that carry health insurance affects access to routine health care and utilization. [31,32] Racial minorities are also less likely to have a primary care provider and are more likely to rely on the Emergency Department for routine care needs. [33-35] As such, radiologists could have an opportunity to help improve healthcare disparities through invention of novel opportunistic imaging screening techniques (i.e., screening an organ for pathology when that organ is incidentally imaged as part of a study obtained for a separate indication). While traditional cataract evaluation will remain the gold-standard for cataract diagnosis, future studies should investigate whether patients presenting to the ED who undergo DE-CT could benefit from opportunistic detection of cataract and a care pathway that refers such patients to ophthalmology for confirmation and treatment.
- The last line on the first page: by “sage-related cataracts” I think the author means age-related cataracts.
We thank the reviewer and have corrected this typographical error.
Reviewer 2 Report
1- I suggest improving the title to something like: "Study of Natural Crystalline Lens Characteristics Using Dual-Energy Computed Tomography"
2-The first sentence of abstract is hard to understand consider changing this sentence: "The relationship between demographic and clinical risk factors for cataract and x-ray at-tenuation of the natural crystalline lens on Dual-Energy Computed Tomography (DECT) is not known."
3- Add a conclusion paragraph at the end of your abstract.
4- The introduction section although well written is somehow too long. Consider shortening it if possible.
5- The discussion section although well written is somehow too long. Consider shortening it if possible.
Author Response
Diagnostics 11/4/22
Dear Ms. Wang,
We thank the editor and reviewers for their thoughtful reviews of our submission and the opportunity to revise the manuscript, with the now revised title “Study of the Natural Crystalline Lens Characteristics Using Dual-Energy Computed Tomography”. Please see our bolded responses to reviewer comments below. As we note, this paper is the first to explore the ability of Dual-Energy Computed Tomography (DECT) to detect differences in the attenuation of the natural crystalline lens between different demographic groups at risk for cataract. While we acknowledge there are limitations when working with a retrospective dataset, we believe the findings are novel and make a meaningful contribution to the field. Moreover, we hope these data will provide the basis of important future prospective work where more definitive ophthalmic evaluation can be proposed in patients undergoing DE-CT of the head during an ED visit. Given the novel imaging findings, we believe this work is well suited to publication in Diagnostics in the special issue Lesion Detection and Analysis using Optical Imaging. Please do not hesitate to reach out if any further questions or requests.
Sincerely,
Jeremy Sachs, MD and Atalie C. Thompson, MD, MPH
Response to Reviewers
Reviewer #2:
1- I suggest improving the title to something like: "Study of Natural Crystalline Lens Characteristics Using Dual-Energy Computed Tomography"
We agree with the reviewer’s suggestion and have changed the title as requested.
2-The first sentence of abstract is hard to understand consider changing this sentence: "The relationship between demographic and clinical risk factors for cataract and x-ray at-tenuation of the natural crystalline lens on Dual-Energy Computed Tomography (DECT) is not known."
We have amended for clarity as requested.
Please see first line of the abstract, lines 13-15:
“There is a paucity of radiologic literature regarding age-related cataract, and little is known about any differences in the imaging appearance of the natural crystalline lens on computed tomography (CT) exams among different demographic groups.”
3- Add a conclusion paragraph at the end of your abstract.
We added a conclusion per request.
4- The introduction section although well written is somehow too long. Consider shortening it if possible.
We thank the reviewer for their suggestion. We have removed some sentences to shorten the introduction.
Please see modified introduction, page 2:
“One reason for this lack of knowledge is that when using conventional single-energy CT techniques, the lenses are bland, relatively homogeneous structures that garner little attention unless they are displaced, dysmorphic, or absent. Moreover, since the natural lens is a radiosensitive organ at particular risk for radiation effects due to its superficial location in the human body, intentional imaging of the lens during radiologic image acquisition is not generally recommended. [9] In clinical practice, however, the lens is often incidentally included in the CT scan field of view. This circumstance presents the radiologist with an opportunity to assess the lens for pathology such as cataracts.
In recent years, dual-energy CT (DECT) has risen to the forefront of CT imaging acquisition due to its ability to capitalize on the differences in energy-dependent x-ray absorption of different materials within the patient by using low- and high-energy x-ray spectra. DECT techniques have numerous recently described applications in the practice of neuroradiology. [10] Low kilo–electron volt (keV) (e.g., 40 keV) virtual monoenergetic imaging (VMI) techniques allow for improved contrast-to-noise ratio among soft tissues of similar attenuation, even in the absence of iodinated contrast media. [11] High keV (e.g., 190 keV) VMI have been used to describe unique attenuation patterns of silicone oil which can be found in the eye after certain retinal surgeries. [12,13]”
5- The discussion section although well written is somehow too long. We have modified and shortened some parts of the discussion.
We thank the reviewer for their suggestion. We have made several modifications in several discussion paragraphs which has led to some reduction in overall length.
For example, please see Discussion, page 8:
“A majority of patients in our study did not have historic ophthalmic data, suggesting there could be gaps in access to ophthalmic care that could be potentially addressed during an emergency department visit. Racial disparities in access to cataract surgery are well-documented, especially among black patients. [30] Inequities in the proportion of patients that carry health insurance affects access to routine health care and utilization. [31,32] Racial minorities are also known to be less likely than white patients to have a primary care provider and are more likely to rely on the Emergency Department for routine care needs. [33-35] As such, radiologists could have an opportunity to help improve healthcare disparities through invention of novel opportunistic imaging screening techniques (i.e., screening an organ for pathology when that organ is incidentally imaged as part of a study obtained for a separate indication). While traditional cataract evaluation will remain the gold-standard for cataract diagnosis, future studies should investigate whether patients presenting to the ED who undergo DE-CT could benefit from opportunistic detection of cataract and development of a care pathway that refers such patients to ophthalmology for confirmation and treatment.
Our study has several limitations. In order to allow for examination of possible age-related differences in the lens, two cohorts of older and younger adults were collected. Thus, the study findings are not generalizable to middle-aged patients. Achieving accurate lens cortex attenuation values is problematic due to the concern for volume averaging with adjacent fluid filled structures. However, if eyes had cortical changes overlying the nucleus these would have been detected within the ROI. Additional ophthalmic clinical data as to the presence or grade of cataract were only available in a subset of older adults. Based on the findings in this study, a future prospective study could be designed in which patients are referred to an ophthalmologist for clinical exam of the lens following incidental imaging of the crystalline lens on DECT during an emergency room encounter.”

Round 2
Reviewer 1 Report
The authors have addressed and clarified most of the comments. Looking forward to the next paper on how they use DECT in cataractous eyes to identify the lens attenuation characteristics and to successfully detect cataract using DECT.
Minor revision to Discussion. The authors mentioned that age-group was not correlated with lens attenuation in multivariable analyses. It was inconsistent with the correlations identified in bivariate analyses and in cataract samples. Can analyze the possible reason in Discussion part?
Author Response
We thank the reviewer for their comment and have added the following to the Discussion, page 7, paragraph 2:
It is possible that DECT did not identify a Hounsfield-unit based threshold to distinguish lens age because different subtypes of cataract may result in different alterations in lens attenuation. For example, some cataracts might decrease lens attenuation if there is an increase in fluid whereas others may increase lens attenuation if there is an increase in deposition of calcium or other high density materials. Future studies could consider whether specific subtypes of cataract have different degrees of lens attenuation on DECT.